 eLIFE

# Symmetric exchange of multi-protein building blocks between stationary focal adhesions and the cytosol

**Jan-Erik Hoffmann[1,2†], Yessica Fermin[3], Ruth LO Stricker[1‡], Katja Ickstadt[3], Eli Zamir[1]\***

[1]Department of Systemic Cell Biology, Max Planck Institute of Molecular Physiology, Dortmund, Germany; [2]Bioanalytics Department, Leibniz Institute for Analytical Sciences, Dortmund, Germany; [3]Faculty of Statistics, TU Dortmund University, Dortmund, Germany

**Abstract** How can the integrin adhesome get self-assembled locally, rapidly, and correctly as diverse cell-matrix adhesion sites? Here, we investigate this question by exploring the cytosolic state of integrin-adhesome components and their dynamic exchange between adhesion sites and cytosol. Using fluorescence cross-correlation spectroscopy (FCCS) and fluorescence recovery after photobleaching (FRAP) we found that the integrin adhesome is extensively pre-assembled already in the cytosol as multi-protein building blocks for adhesion sites. Stationary focal adhesions release symmetrically the same types of protein complexes that they recruit, thereby keeping the cytosolic pool of building blocks spatiotemporally uniform. We conclude a model in which multi-protein building blocks enable rapid and modular self-assembly of adhesion sites and symmetric exchange of these building blocks preserves their specifications and thus the assembly logic of the system.

**\*For correspondence:** eli.zamir@mpi-dortmund.mpg.de

**Present address:** †Cell Biology and Biophysics Unit, European Molecular Biology Laboratory, Heidelberg, Germany; ‡Department of Infection Genetics, Helmholtz Centre for Infection Research, Braunschweig, Germany

**Competing interests:** The authors declare that no competing interests exist.

**Reviewing editor**: Robert H Singer, Albert Einstein College of Medicine, United States

## Introduction

The integrin adhesome contains more than hundred proteins with multiple binding sites for each other (*Zamir and Geiger, 2001*; *Zaidel-Bar et al., 2007*; *Byron et al., 2011*; *Kuo et al., 2011*; *Schiller et al., 2011*; *Geiger and Zaidel-Bar, 2012*). Upon local signals, components of the integrin adhesome get self-assembled along the plasma membrane to form nascent adhesion sites and to further mature them to focal complexes and focal adhesions (*Zamir and Geiger, 2001*; *Cukierman et al., 2002*; *Gardel et al., 2010*; *Zaidel-Bar and Geiger, 2010*). These adhesion sites assemble, disassemble, and change their molecular content by exchanging their components with the surrounding cytosol (*Wolfenson et al., 2011*; *Lavelin et al., 2013*). Likewise, stationary adhesion sites are dynamically maintained by a balanced rapid exchange of constituents with the cytosol (*Lele et al., 2008*; *Wolfenson et al., 2011*; *Lavelin et al., 2013*). Considering the molecular diversity and complexity of cell-matrix adhesion sites (*Zamir et al., 1999*; *Zamir and Geiger, 2001*; *Cukierman et al., 2002*; *Zamir et al., 2008*), a major question is how can the integrin adhesome form these structures rapidly, locally, and correctly. In order to approach this question it is essential to consider and examine not only the adhesion sites themselves but also the state of their components in the cytosolic pool. However, the study of this fundamental aspect was largely abandoned so far, due to tacitly conceiving the cytosolic pool as a passive supply of individual integrin adhesome proteins. In this work we challenged this view by systematic FCCS and FRAP measurements. Strikingly, we found that the integrin adhesome is actually extensively pre-assembled already in the cytosol, forming multi-protein building blocks that can facilitate rapid and modular assembly of adhesion sites. The physical associations between adhesion site

**eLife digest** The space between the cells in a multi-cellular organism is filled with a structure called the extracellular matrix. The roles of this matrix vary from tissue to tissue, and include providing the cell with structural support and signalling cues.

Cells use large multi-protein structures along their plasma membrane, called cell-matrix adhesion sites, to attach to the extracellular matrix and sense its properties. These sites are rapidly self-assembled upon local cues and are dynamically maintained by exchanging their components with the cytosol of the cell. Given the large number of different proteins that can be used to build adhesion sites, and the large number of possible interactions between these proteins, a major question is—how are these structures self-assembled both rapidly and correctly?

Most research so far has concentrated on the adhesion sites themselves, as it has generally been assumed that their constituent proteins enter and leave these sites individually. However, using fluorescence techniques, Hoffmann et al. have now shown that these proteins are actually pre-assembled in the cytosol into small 'building blocks'. These can make the construction of cell-matrix adhesion sites quicker and more accurate by reducing the number of steps in the self-assembly process. In addition, under most circumstances, these building blocks do not change when they are exchanged between the cytosol and the adhesion site: this means that they can be re-used repeatedly and accurately. The work of Hoffmann et al. also raises questions about whether similar principles govern the self-assembly of other large intracellular multi-protein structures.

components are spatially uniform around stationary focal adhesions, indicating that these sites exchange their building blocks with the cytosol without altering them (i.e., symmetrically). Together, our results lead to a concept in which self-assembly of adhesion sites involves recruitment of multi-protein modules and symmetric exchange preserves the specifications of these building blocks in the cytosolic pool, thereby facilitating correct self-assembly.

## Results

We first asked whether the integrin adhesome forms in the cytosol pre-assembled multi-protein building blocks for adhesion sites. To address this question we selected 13 key components of adhesion sites—α-actinin, α-parvin, p130CAS (CAS), CSK, FAK, ILK, paxillin, PINCH, talin, tensin, VASP, vinculin, and zyxin—and quantified systematically all 91 possible pairwise physical associations between them in the cytosol of REF52 cells using FCCS (*Figure 1*). The sensitivity of FCCS enabled to perform these measurements at low concentrations of ectopically expressed proteins, comparable to the range of typical endogenous concentrations (*Figure 1—figure supplement 1*). The obtained data was analyzed to derive for each measurement the association score, which weights the cross-correlation amplitude by its noise, and the apparent $Ka$ that considers also the auto-correlation curves to quantify the apparent association strength ('Materials and methods'). The overall distribution of the association scores was skewed to the right as compared to the negative-control measurements (*Figure 1B*, top), indicating the presence of physical associations between some of the analyzed proteins in the cytosol. These positive measurements resulted from a subset of the 91 protein pairs (*Figure 1B*). Among the 91 possible pairing combinations, 18 protein pairs were found to be physically associated with a p-value <0.0001 (*Figure 1C,D*; *Supplementary file 1*). Among these pairs, 15 pairs were found to be physically associated also in the cytosol of another cell line, NIH3T3, with a p-value <0.02 (*Figure 1E*; *Supplementary file 1*). These results show a strikingly extensive degree of physical associations between adhesion site components in the cytosol. This indicates that the assembly of adhesion sites is plausibly carried out by recruitment of pre-assembled multi-protein building blocks, rather than of individual proteins.

A network of proteins that have multivalent interactions with each other has the potential to form large, high-order, complexes. Therefore we sought to assess the size of protein complexes formed by the integrin adhesome in the cytosol. If a given high-order protein complex exists, then each pair of its components is expected to exhibit a pairwise physical association. Accordingly, by searching for fully intra-connected sub-graphs in the measured network of pairwise physical associations (*Figure 1E*), two potential ternary complexes can be inferred: an ILK-PINCH-α-parvin complex, which was indeed

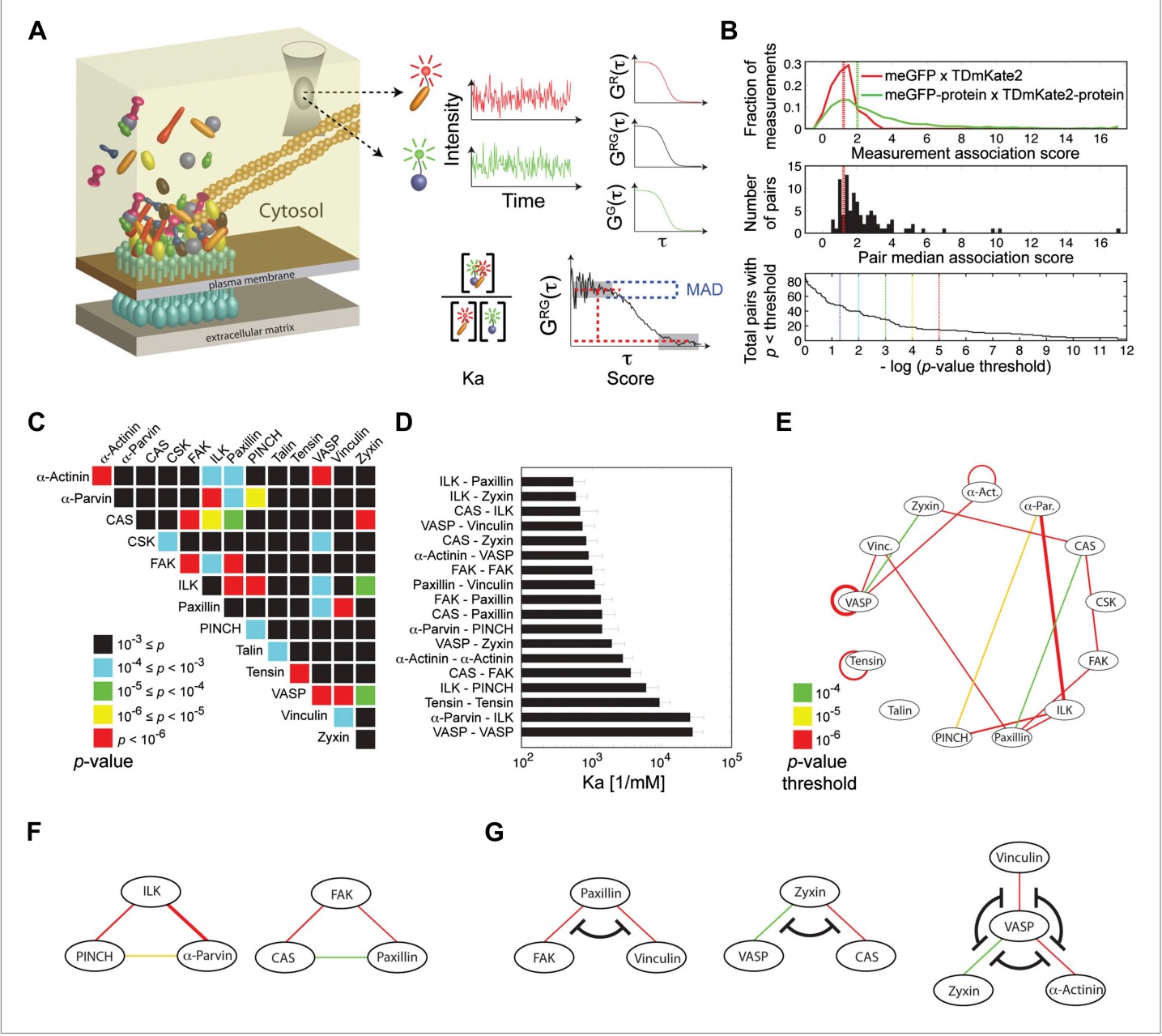

**Figure 1**. Extensive physical associations between components of cell-matrix adhesion sites in the cytosol. (**A**) Pairwise physical associations between proteins tagged with meGFP and TDmKate2 were measured in the cytosol of REF52 and NIH3T3 cells using FCCS (schematized). From these measurements the apparent association constants (*Ka*) and the association scores were derived as described. (**B**) Top, area-normalized distributions of association scores between meGFP and TDmKate2 alone (i.e., negative control, n = 126 cells, red line) and between the different analyzed components of adhesion sites in all individual valid measurements performed in REF52 cells (n = 1914 cells; green line) with their corresponding medians (vertical lines). Middle, the distribution of median association score of the 91 protein pairs (60 ≥ n ≥ 9 cells per each pair). Red line indicates the median association score of the negative control. Bottom, the total number of pairs with a median association score bigger than that of the negative control at different statistical confidences (***Supplementary file 1***). The p-values indicate the probability that the observed median association score of a given pair is bigger than that of the negative control by coincidence. Thus a higher −log(p-value threshold) value means a higher statistical confidence for physical association. (**C**) A heatmap indicating the p-value of each protein pair in REF52 cells. (**D**) A bar plot showing the median ± median absolute deviation (MAD) *Ka* for protein pairs having p-value <0.0001 (n ≥ 13 cells per pair). (**E**) The network of physical associations between the analyzed proteins. Shown edges are those having p-value <0.0001 in REF52 and p-value <0.02 in NIH3T3 (***Supplementary file 1***). Edges color and width indicate p-value categories as in (**C**) and proportionally *Ka* in REF52, respectively. (**F**) Based on the network shown in (**E**), two potential ternary complexes are

*Figure 1. Continued on next page*

*Figure 1. Continued*

indicated. (**G**) Mutually exclusive physical associations inferred from (**D**) and (**E**) as cases in which two or more proteins exhibit pairwise associations with another protein but not between themselves.

The following figure supplements are available for figure 1:

**Figure supplement 1**. Concentrations of the ectopically expressed proteins in the FCCS measured cells.

reported as a ternary complex (**Wu, 2004**; **Legate et al., 2006**), and a CAS-FAK-paxillin complex (**Figure 1F**). Notably, there is no complex containing more than three of the analyzed proteins. This indicates that despite the abundant multivalent interactions in the integrin adhesome network, its cytosolic complexes are considerably confined due to mutual-exclusiveness between protein interactions and allosteric regulations. For example, paxillin was found to be physically associated with vinculin and FAK, however, no association was found between vinculin and FAK (**Figure 1E,G**). This suggests that the associations of vinculin and FAK with paxillin are mutually exclusive and therefore a ternary FAK-paxillin-vinculin complex cannot be formed. This inferred mutual exclusive relation is consistent with studies reporting that common sites along paxillin mediate its interactions with FAK and vinculin (**Turner and Miller, 1994**; **Brown et al., 1996**). Similarly, our results suggest mutual exclusiveness between the associations of VASP with zyxin, α-actinin, and vinculin and between the associations of zyxin with VASP and CAS (**Figure 1G**). We postulate that the size-confinement of cytosolic complexes reflects a system-level design of interdependencies between protein interactions throughout the integrin adhesome network to prevent stochastic assembly of aberrant large complexes.

To assess the diversity of integrin adhesome protein complexes that indeed serve as building blocks for adhesion sites, we asked whether each analyzed protein is a component of only one type of building block (**Figure 2A**). In such a case, each two physically associated proteins should exhibit similar dwell time and mobile fraction in adhesion sites, since they are part of only one, same, type of building block (**Figure 2A**). To test this, we measured the mobile fraction and dwell time ($\tau_{1/2}$) of the 13 proteins in focal adhesions by FRAP (**Figure 2B,C,E,F**) and quantified the difference in dynamics (co-dynamics distance) between each two proteins (**Figure 2G**). Plotting these co-dynamics distances vs the association scores indicated some protein pairs as prominent building blocks (**Figure 2H**), including ILK-α-parvin, ILK-PINCH and α-parvin-PINCH (reflecting the trimetric ILK-PINCH-α-parvin complex) (**Wu, 2004**; **Legate et al., 2006**), paxillin-vinculin, CAS-FAK, VASP-zyxin, and α-actinin-VASP (**Figure 2H**). Noteworthy, the proteins of each of these pairs have also similar diffusion speeds, as measured by FCS (**Figure 2B,D**, **Figure 2—figure supplement 1**), suggesting that they are predominantly bound to each other, rather than with other proteins, in the cytosolic pool. However, besides the abovementioned exceptions, our results indicate that in general physical associations in the cytosol do not necessarily imply similarity in dynamics in adhesion sites. Therefore, we conclude that the cytosolic building blocks of adhesion sites are combinatorially diversified and that most proteins can be recruited to these sites as part of different types of building blocks.

The diversity of the cytosolic building blocks triggers the question whether they are functional and structural modules in adhesion sites. Some of the analyzed proteins can be classified into three groups based on their function and their reported vertical position across focal adhesions (**Kanchanawong et al., 2010**). Accordingly, FAK and paxillin are located in the integrin-signalling layer, vinculin is in the force transduction layer and zyxin, VASP, and α-actinin are in the actin regulation layer (**Kanchanawong et al., 2010**) (**Figure 2I**). Superimposing this information with the physical associations found in the cytosol (**Figure 1E**) shows that such associations occur only between proteins from the same functional group and vertical layer, or between proteins from adjacent vertical layers without including any other analyzed protein in the complex. For example, paxillin and vinculin are physically associated in the cytosol but they do not contain FAK or VASP in their common building block (**Figure 2I**). Therefore, the cytosolic building blocks appear to be modules that are continuous along the functional axis of anchoring-mechanosensing-actin regulation and along the vertical axis across focal adhesions.

Cell-matrix adhesion sites are dynamically maintained structures that constantly exchange their constituting material with the cytosol (**Lele et al., 2008**; **Wolfenson et al., 2009**). We pose here two alternative models for the mode of this exchange—symmetric and asymmetric (**Figure 3A**). In a symmetric exchange, proteins exit from adhesion sites with the same state they had upon entering these

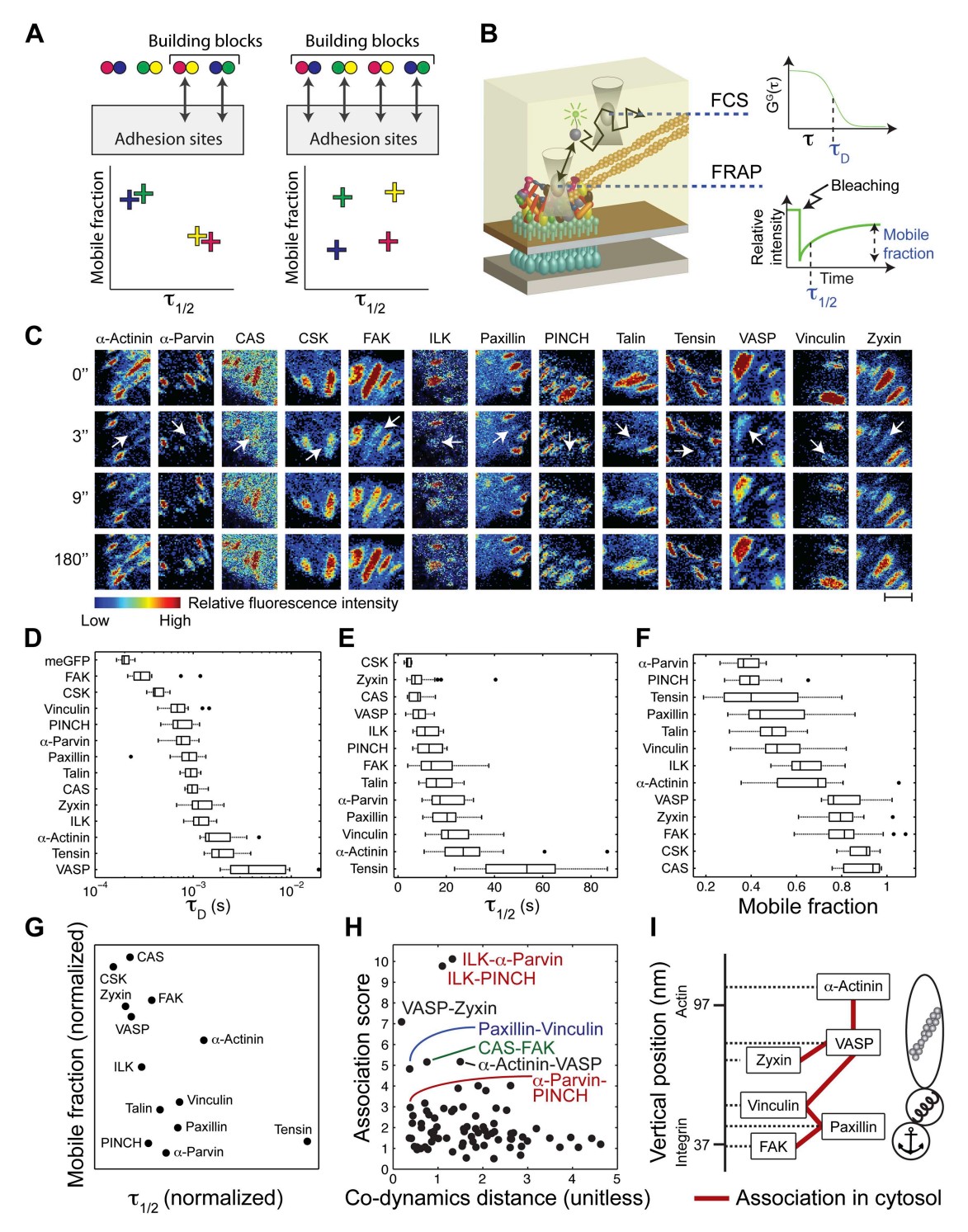

**Figure 2**. The cytosolic building blocks of cell-matrix adhesion sites are combinatorially diversified. (**A**) Not all cytosolic integrin-adhesome complexes are necessarily building blocks for adhesion sites. If each protein is in only one type of building blocks then physically associated proteins should exhibit the same dwell times ($\tau_{1/2}$) and mobile fractions in focal adhesions. (**B**) REF52 cells expressing the analyzed proteins tagged with meGFP were measured using FRAP and FCS to quantify their $\tau_{1/2}$ and mobile fractions in focal adhesions and their dwell times in a confocal volume in the cytosol ($\tau_D$). (**C**) Example FRAP images before (0") and after bleaching a focal adhesion (arrows). Scale bar, 5 μm. (**D–F**) Box plots of the $\tau_D$ (26 ≥ $n$ ≥ 14 cells), $\tau_{1/2}$ and mobile fractions (31 ≥ $n$ ≥ 7 cells) of each protein. (**G**) Median $\tau_{1/2}$ vs median mobile fraction of each protein normalized to zero-mean and unit-variance. Thus, in this plot the Euclidean distance (co-dynamics distance) between proteins quantifies the difference in their dynamics. (**H**) The co-dynamics

*Figure 2. Continued on next page*

*Figure 2. Continued*

distance vs median association score of all possible 78 heteromeric protein pairs. (**I**) The reported vertical distance from substrate of 6 of the analyzed components across focal adhesions (*Kanchanawong et al., 2010*) and the cytosolic associations between them as measured here (*Figure 1E*). Anchor, spring, and actin symbols indicate vertical layers of integrin signalling, mechanosensing and actin regulation across focal adhesions, respectively (*Kanchanawong et al., 2010*).
The following figure supplements are available for figure 2:

**Figure supplement 1**. The relation between physical associations and similarity in diffusion speeds in the cytosol.

sites, while in an asymmetric exchange they exit in a different state, termed herein a primed state. This priming can be any change in the post-translational modifications or interactions of the exiting components. The primed state is expected to be unstable in the cytosolic environment, since otherwise it would have been the characteristic state of the protein in the pool. Therefore, an asymmetric exchange, but not a symmetric exchange, would generate a spatial gradient of primed components emanating from adhesion sites. Noteworthy, both symmetric and asymmetric exchanges allow for a spatiotemporal steady state between adhesion sites and cytosol. However, only an asymmetric exchange enables one adhesion site to affect another one, or itself, via the primed components that it releases to the cytosol. An asymmetric exchange would also generate a locally heterogeneous cloud of building blocks around adhesion sites, which may pose an engineering challenge for ensuring correct self-assembly and maintenance of these sites. In contrast, a symmetric exchange ensures a spatially uniform pool of building blocks for adhesion sites and prevents any communication between these sites via primed components.

In order to experimentally discriminate between symmetric and asymmetric exchanges of protein complexes, we measured the physical associations between integrin adhesome components at spots near (<1.5 μm) and far from focal adhesions (*Figure 3*). We examined this for 28 protein pairs, representing the wide range of association scores exhibited by the 91 protein pairs. For all the 28 protein pairs we found no significant difference in association scores near and far from focal adhesions, supporting the model of symmetric exchange of protein complexes (*Figure 3B–F*; *Supplementary file 1*). The symmetric exchange model also predicts that even when proteins are directly interacting in adhesion sites, they will not be associated close to adhesion sites unless they are associated also far away from adhesion sites (*Figure 3B*). Combining the FCCS measurements with fluorescence lifetime imaging microscopy (FLIM) of four protein pairs, we found indeed that the physical associations near focal adhesions are correlated with their associations far from focal adhesions regardless of their interaction state in focal adhesions (*Figure 3G,H*). These results indicate that there are no spatial gradients of different (i.e., primed) protein complexes around focal adhesions, indicating that stationary focal adhesions release the same types of complexes that they recruited.

Potential priming of a protein in adhesion sites might also be based on a change in its phosphorylation, rather than in its interactions (*Figure 3I*). Therefore, we further tested whether tyrosine-phosphorylated proteins in adhesion sites, such as FAK, paxillin, and CAS (*Figure 3—figure supplement 1*) are recruited and released in different phosphorylation states. To monitor tyrosine phosphorylation in live cells we used a biosensor consisting of two SH2 domains, dSH2, tagged with a fluorescent protein (*Kirchner et al., 2003*). This reporter is enriched in focal adhesions (*Kirchner et al., 2003*), where it also exhibits FRET with FAK, paxillin, and CAS (*Figure 3—figure supplement 1*). The association scores of all three proteins with dSH2 show no significant difference between the near and far locations, indicating that there is no detectable spatial gradient in the phosphorylation level of paxillin, CAS, and FAK around focal adhesions (*Figure 3J,K*). Noteworthy, inhibition of phosphatases by vanadate increases the association scores between dSH2 and CAS, paxillin and FAK, indicating the capability of the assay to detect changes in phosphorylation levels of proteins in the cytosol (*Figure 3L*). Together, these results support the model of symmetric material exchange for stationary focal adhesions. Accordingly, stationary focal adhesions do not alter the cytosolic pool of building blocks and do not communicate or interfere with nearby adhesion sites via primed components.

## Discussion

We conclude that the integrin adhesome is extensively pre-assembled in the cytosol, thereby forming multi-protein building blocks for cell-matrix adhesion sites. Our results further indicate that these

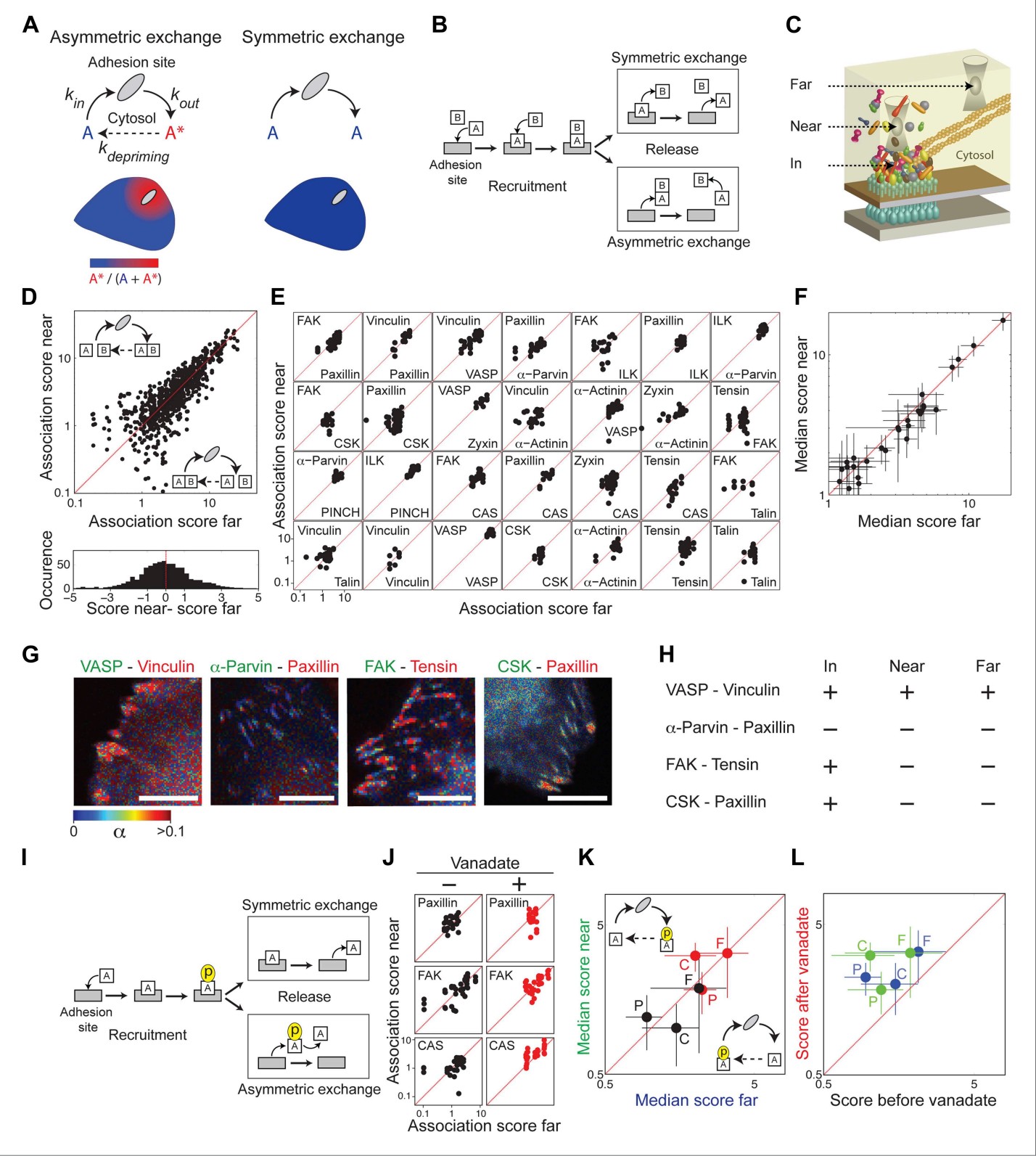

**Figure 3**. Symmetric material exchange between stationary focal adhesions and cytosol. (**A**) The symmetric and asymmetric models of material exchange between adhesion sites and cytosol. In symmetric exchange a component, *A*, exits from adhesion site in the same state it had upon entering to it. In asymmetric exchange *A* exits in a different, primed state *A\** and relaxes back to state *A* in the cytosol, thereby generating a spatial gradient of the primed state emanating from adhesion sites. (**B**) Formulation of the two models for the case in which priming (of *A*) is based on interaction

*Figure 3. Continued on next page*

*Figure 3. Continued*

(with protein *B*). Here, asymmetric exchange would generate a spatial gradient of *AB* complex concentration emanating from adhesion sites. (**C**–**F**) Discriminating between the two models by measuring the physical associations near (<1.5 µm) and far from focal adhesions for 28 protein pairs as named in (**E**). Scatter plots compare the association scores near vs far from focal adhesions for all the 28 pairs together (*n* = 755 focal adhesions) (**D**), for each pair separately (*n* ≥ 9) (**E**) or for the median score of each pair ± MAD (**F**). Data-points far from the equality diagonals (dashed red lines) would correspond to asymmetric exchanges, as illustrated in (**D**), while data-points along the diagonal indicate symmetric exchange. Histogram in (**D**) shows the distribution of the difference in association scores near and far from focal adhesions. (**G**) FLIM images color-coding the fraction, α, of donor- (mCitrine-) tagged protein (green) that FRETs to the acceptor- (mCherry-) tagged protein (red) for four protein pairs. Scale bars, 10 µm. (**H**) Comparison of the interaction states of the four protein pairs shown in (**G**) in focal adhesions with their physical associations near and far from focal adhesions. (**I**) Formulation of the symmetric and asymmetric models for the case in which priming is based on phosphorylation. (**J**–**L**) Scatter plots comparing the association scores of meGFP-dSH2 with paxillin, FAK, and CAS (denoted P, F, and C, respectively) near vs far from focal adhesions (green and blue, respectively) and before vs after vanadate treatment (black and red, respectively). Error bars indicate MAD (*n* ≥23 focal adhesions).

The following figure supplements are available for figure 3:

**Figure supplement 1**. FAK, paxillin, and CAS are tyrosine-phosphorylated and interact with SH2 domain in focal adhesions.

building blocks are size-confined (***Figure 1***) and are predominantly corresponding to the functional layers across focal adhesions of anchoring and integrin regulation, force transduction, and actin regulation (***Figures 2 and 4***). In the cytosol, the building blocks cannot form bigger structures due to dependencies between protein interactions. We propose that at the plasma membrane the system can get locally switched on to self-assemble adhesion sites by passing through checkpoints that induce additional protein interactions in the integrin adhesome (***Figure 4***). These checkpoints include mechanical stabilization of the active conformation of matrix-anchored integrin by actin polymerization (***Puklin-Faucher and Sheetz, 2009***; ***Vicente-Manzanares et al., 2009***; ***Gardel et al., 2010***; ***Shattil et al., 2010***), activation of proteins (e.g., vinculin) at the plasma membrane by phosphatidyl-inositol-4-5-bisphosphate (PIP2) (***Gilmore and Burridge, 1996***; ***Zamir and Geiger, 2001***), and mechanical stretching of proteins (e.g., talin and CAS) by actomyosin contractility (***Riveline et al., 2001***; ***Sawada et al., 2006***; ***Johnson et al., 2007***; ***del Rio et al., 2009***; ***Puklin-Faucher and Sheetz, 2009***; ***Gardel et al., 2010***). Since adhesion sites are self-assembled structures, a standardized pool of building blocks is essential for their correct and efficient assembly. Symmetric material exchange between adhesion sites and cytosol satisfies this requirement by ensuring that building blocks will enter and exit adhesion sites without being altered.

Interestingly, in contrast to stationary focal adhesions, disassembling focal-adhesions might exchange material with the cytosol in an asymmetric manner. A previous study reported that in mouse embryonic fibroblasts FAK and vinculin associate with paxillin very close to disassembling focal adhesions but not far away from them (***Digman et al., 2009***). This study was based on a spatially resolved FCCS technique that gains spatial information on the expense of the sampling rate and integration time spent on each location, hence on the expense of the signal-to-noise ratio. This might underlie the difference from our results, which show the presence of FAK-paxillin and vinculin-paxillin associations far away from focal adhesions in REF52 and NIH3T3 cells. Yet, this study (***Digman et al., 2009***) importantly indicates a gradient in association strengths between these proteins emanating from disassembling focal adhesions, supporting in this case the asymmetric material exchange model. Asymmetric material exchange by disassembling focal adhesions can enable their rapid destruction by breaking them up to relatively bigger pieces, as well as be a mechanism by which they can affect other adhesion sites.

The results presented here revise the concept about adhesion sites self-assembly, from a process that is mediated by the recruitment of individual proteins to a process involving pre-assembled multi-protein building blocks. This has implications for broad aspects of cell-matrix adhesion sites formation and regulation. Multi-protein building blocks, in comparison to single-protein building blocks, reduce the number of steps required for self-assembly of adhesion sites. With a properly designed regulation, such a reduction of steps can make the self-assembly of adhesion sites less sensitive to stochastic biochemical noise and therefore also faster. The regulation of adhesion sites assembly should cope with the combinatorial diversity of the building blocks. Since each protein can be a component of different building blocks, this regulation should consider each building block as a whole rather than its constituting proteins as individuals. Otherwise, the self-assembly process would lose control on the identity of the co-recruited proteins when recruiting a protein embedded in a multi-protein building block and

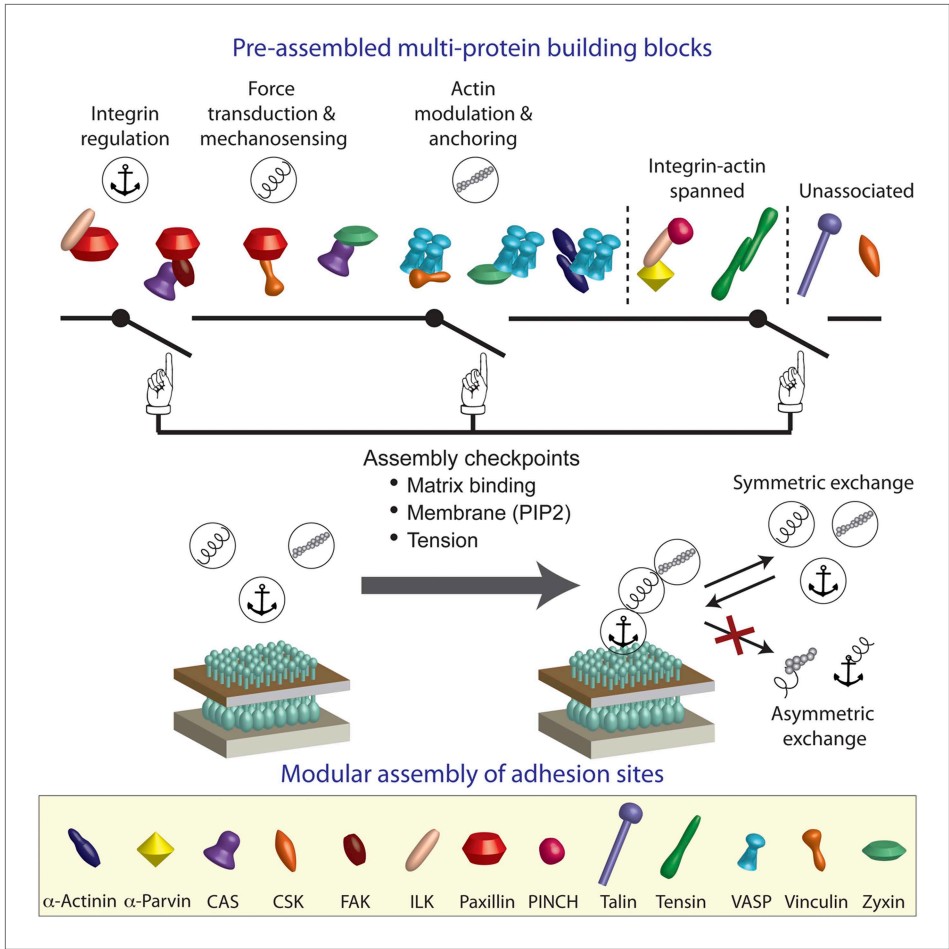

**Figure 4**. A model of switchable formation of adhesion sites via pre-assembled multi-protein building blocks. The integrin adhesome is pre-assembled in the cytosol as multi-protein building blocks for adhesion sites. These building blocks are combinatorially diversified but confined in their size. Most of the building blocks form modules that are consistent with the previously reported (*Kanchanawong et al., 2010*) vertical continuum of anchoring, mechanosensing, and actin regulation layers across focal adhesions. In the cytosol, the pre-assembled building blocks cannot further assemble to form bigger structures due to mutual exclusiveness between protein interactions and allosteric regulations. On the plasma membrane, the system can get locally switched on to assemble an adhesion site by passing through checkpoints that enable additional protein interactions in the integrin adhesome. These checkpoints include anchoring of integrins to the extracellular matrix, mechanical stretching of proteins like talin and CAS by actomyosin contractility, and activation of proteins like vinculin and talin by PIP2 on the plasma membrane. Symmetric material exchange between adhesion sites and cytosol retains the wiring of the building blocks and therefore retains the assembly logic and switchability of the system.

thereby will be drifted to form aberrant structures. In order to understand such assembly logic of adhesion sites, it would be important to integrate interaction-dependencies across the integrin adhesome (*Köster et al., 2012*) and to comprehensively characterize the diversity of its high-order protein complexes in the cytosol by combining advanced optical and proteomic approaches (*Heinze et al., 2004*; *Jain et al., 2011*; *Ridgeway et al., 2012*).

# Materials and methods

## Cells and transfections

Rat embryonic fibroblasts, REF52 (kindly provided by Joachim Spatz and Benjamin Geiger) and NIH3T3 cells were maintained in culture in Dulbecco's modified Eagle's medium (DMEM; PAN-Biotech, Aidenbach, Germany, 4.5 g/l glucose) supplemented with 10% FBS (PAN-Biotech), 2 mM Gln, and

1% non-essential amino acids. 2 days prior experiments, cells were plated in MatTek dishes (MatTek Corporation, Ashland, MA), incubated for 24 hr, and then transfected with 0.4 µg of each plasmid as indicated using Lipofectamine 2000 (Invitrogen, Carlsbad, CA). Shortly before experiments growth medium was replaced with 2 ml HEPES buffered imaging medium (PAN-Biotech).

## Fluorescence correlation/cross-correlation spectroscopy (FCS/FCCS)

FCCS measurements were conducted on a Zeiss LSM 510 META ConfoCor 3 System (Carl Zeiss AG, Oberkochen, Germany) using a 40x 1.2 NA water objective, at 37°C. Cells expressing meGFP and TDmKate2 tagged proteins were excited simultaneously with 488 nm and 594 nm laser lines passing through 405/488/594 HFT. Emission light was split with a 565 NFT, further filtered via BP 505-540 and LP 655 filters for the green and red channels, respectively, and collected by the internal APDs. For each measurement 10 traces of 10 s each were sequentially acquired. FCS/FCCS calibration and control measurements were performed as explained in the FCCS data analysis section. For inhibition of phosphatases, cells were treated with 100 µM sodium ortho-vanadate (vanadate) for 30 min. The data of each FCCS measurement was analyzed to obtain the apparent association constant ($Ka$) and association score as explained in the FCCS data analysis section. Statistical tests of significance were performed as described in the Statistical analysis section and *Supplementary file 1*. Dwell times of proteins in the confocal volumes, $\tau_D$, were measured by FCS in REF52 cells expressing these proteins tagged with meGFP using the same setting as for FCCS but without the 594 nm laser line excitation. The autocorrelation curves of these FCS data were then fitted to a single component diffusion model:

$$G(\tau) = \frac{1}{N} \cdot \frac{1}{\left(1 + \dfrac{\tau}{\tau_D}\right)\sqrt{1 + \dfrac{\tau}{s^2 \cdot \tau_D}}} + G_\infty \qquad (1)$$

where $s$ is the structural parameter, $G_\infty$ is the offset, and $N$ is the average number of particles in the confocal volume, in order to derive $\tau_D$.

## Fluorescence recovery after photobleaching (FRAP)

FRAP measurements were performed on a Zeiss LSM 510 META using a 40x 1.2 NA water objective at 37°C. REF52 cells expressing meGFP-tagged proteins were excited with a 488 nm laser line (10% intensity) passing through a 405/488/594 HFT. Emission light was split with a 565 NFT and further filtered with a BP 505-550 using an 896 µm (12.5 airy units) pinhole. Images were acquired with 3 s intervals throughout the measurements. After the tenth frame, the fluorescence of a complete, single focal adhesion was bleached by 30 pulses of 100% laser intensity over a period of 1–2 s and image acquisition continued for a period of at least 180 s. The bleaching of a complete adhesion sites was done in order to ensure that the fluorescence recovery is due to exchange of material between adhesion sites and cytosol and not with unbleached parts within the adhesion site itself. Therefore, in different FRAP experiments the size of the bleached area was different. However, per protein we observed no correlation between this area and the recovery parameters, with the exception of CSK and zyxin. Image analysis and data analysis of the FRAP data were performed using ImageJ (US National Institutes of Health, Bethesda, MD; http://imagej.nih.gov/ij/) and Matlab (Mathworks, Natick, MA), respectively. The intensity recovery trace of each bleached focal adhesion was quantified and subtracted by a corresponding background trace measured within the same cell, far from the bleached area. Then, each curve was normalized in respect to the intensity levels before bleaching and upon bleaching, $I(t) = (I^{un}(t) - I^{un}_{bleach})/(I^{un}_{pre} - I^{un}_{bleach})$, where $I^{un}$ is the background-subtracted intensity of the focal adhesion before normalization, $I^{un}_{pre}$ is the mean value of $I^{un}$ in the nine time points before bleaching, and $I^{un}_{bleach}$ is the value of $I^{un}$ at the time point immediately after bleaching. Each derived curve was then fitted to a mono-exponential recovery model, $I(t) = M(1 - e^{-\tau \cdot t})$, using the Matlab fit function (with the method being set to Nonlinear Least Squares), where $M$ is the mobile fraction and $\tau$ was used to derive the dwell time, $\tau_{1/2}$ as $\tau_{1/2} = -\ln(0.5)/\tau$. Curves that yielded a fit with a coefficient of determination (R-square) ≤0.7 were excluded from further analysis.

## Fluorescence lifetime imaging microscopy (FLIM)

FLIM experiments were performed on an Olympus FluoView 1000 laser scanning microscope (Olympus, Tokyo, Japan) with a Sepia II and PicoHarp 300 time correlated single photon counting system from

PicoQuant (Berlin, Germany). Proteins were tagged with the fluorescent proteins mCitrine (donor) and mCherry (acceptor) to optimize FRET efficiency. All experiments were carried out at 37°C with a 60x water objective. For imaging, samples were simultaneously excited with 488 (10% laser intensity) and 561 nm (31% laser intensity) light using a 405/488/561/633 dichroic mirror. Excitation light was split with a dichroic mirror at 560 nm and detected between 500–550 nm (green channel) and 580–680 nm (red channel). The pinhole was set to 300 μm. FLIM images were acquired using 470 nm excitation (36% intensity) with a pulse frequency of 40 MHz, a 405/470 dichroic mirror, and a 525/15 band path filter. FLIM images were analyzed using IGOR Pro (Version 6.22 A, Wave Metrics, Lake Oswego, OR) with pFLIM3 as previously described (*Walther et al., 2011*).

## FCCS data analysis

Definitions:

$V_R$ = The size of the red volume
$V_G$ = The size of the green volume
$V_{RG}$ = The size of the overlapping volume
$G_0^R$ = The amplitude of the red auto-correlation curve
$G_0^G$ = The amplitude of the green auto-correlation curve
$G_0^{RG}$ = The amplitude of the cross-correlation curve
$N_R^R$ = The number of all red particles in the red volume
$N_G^G$ = The number of all green particles in the green volume
$N_{RG}^{RG}$ = The number of all red-green complexes in the overlapping volume
$[R_T]$ = The concentration of all red particles
$[G_T]$ = The concentration of all green particles
$[RG]$ = The concentration of all red-green complexes
$[R]$ = The concentration of unbound red particles
$[G]$ = The concentration of unbound green particles
$C_R$ = The count rate in the red channel
$C_G$ = The count rate in the green channel
$C_D$ = The dark count rate of the detectors

The acquired fluorescence fluctuation traces were correlated using the Zeiss ConfoCor 3 software to derive their corresponding auto-correlations and cross-correlation curves (*Equation 2*),

$$G^{xy}(\tau) = 1 + \frac{\langle \delta I_x(t) \cdot \delta I_y(t+\tau) \rangle}{\langle I_x \rangle \langle I_y \rangle}$$

(2)

where $x$ and $y$ denote the correlated channels ($R$ and $G$ for the TDmKate2 and meGFP channels, respectively, $\langle \ \rangle$ denotes mean and $\delta I(t) = I(t) - \langle I(t) \rangle$).

The amplitudes of the FCCS auto-correlation curves ($G_0^{RR}$ and $G_0^{GG}$ for the red and green auto-correlation curves, respectively) and cross-correlation curve ($G_0^{RG}$) were calculated as the median value of each curve between 25.6 μs and 81.92 μs (a range containing 41 logarithmically spaced data points) subtracted with the median value of the offset as sampled on that curve from 157.3 ms to 838.9 ms (a range containing 21 logarithmically spaced data points). In contrast to the auto-correlation curves, if the two measured proteins are not physically associated the cross-correlation curve might get slightly negative values. Such cross-correlation amplitudes values were set then to zero, to capture the information that no physical association was detected, while allowing to batch calculations of Ka for all measurements together.

Based on $G_0^{RR}$ and $G_0^{GG}$, the average number of TDmKate2 particles and meGFP particles in the corresponding confocal volumes were calculated: $N_R^R = 1/G_0^R$, $N_G^G = 1/G_0^G$, $N_{RG}^{RG} = G_0^{RG} N_R^R N_G^G$. Dividing these particle numbers with the corresponding confocal volumes provides the total concentrations of the particles: $[R_T] = N_R^R/V_R$, $[G_T] = N_G^G/V_G$, $[RG] = N_{RG}^{RG}/V_{RG}$, where $[R] = [R_T] - [RG]$ and $[G] = [G_T] - [RG]$. Accordingly, the apparent association constant, Ka, equals:

$$Ka = \frac{[RG]}{[R][G]} = \frac{N_{RG}^{RG}/V_{RG}}{\left(N_R^R/V_R - N_{RG}^{RG}/V_{RG}\right)\left(N_G^G/V_G - N_{RG}^{RG}/V_{RG}\right)}.$$

(3)

The values of the effective volumes ($V_{eff}$) of $V_R$ and $V_G$ were derived as previously described (**Rüttinger et al., 2008**), by approximating the confocal volume, $V_{conf}$, to a 3D Gaussian shape (**Equation 4**) with radial radius $\omega_0$, axial radius $z_0$, and $s = z_0/\omega_0$ (the structural parameter):

$$V_{conf} = \left(\frac{\pi}{2}\right)^{3/2} \cdot \omega_0^2 \cdot z_0 = \left(\frac{\pi}{2}\right)^{3/2} \cdot \omega_0^3 \cdot s = \left(\frac{1}{2}\right)^{3/2} \cdot V_{eff}, \tag{4}$$

and therefore:

$$V_{eff} = \pi^{3/2} \cdot \omega_0^2 \cdot z_0. \tag{5}$$

In order to measure $\omega_0$ and $s$, FCS measurements of green and red dyes (Oregon green and Atto 655-maleimide, respectively) with known diffusion coefficients (in 25°C: $4.11\cdot10^{-6}$ cm²/s and $4.09\cdot10^{-6}$ cm²/s, respectively) were performed. Based on **Equation 6**, the diffusion coefficients for Oregon green and Atto 655-maleimide at measurement temperature, 37°C, can be calculated (**Dross et al., 2009**):

$$D(T_2) = D(T_1) \cdot \frac{T_2}{T_1} \cdot \frac{\eta(T_1)}{\eta(T_2)}, \tag{6}$$

where $T_1$ and $T_2$ are the compared temperatures (298.15 K and 310.15 K, respectively), $D$ the diffusion coefficient (cm²/s), and $\eta$ the viscosity of water (Pa·s) at these temperatures. The viscosity of water was calculated using **Equation 7** (**Dross et al., 2009**) with values A = $2.414\cdot10^{-5}$ Pa·s, B = 247. 8 K, and C = 140 K, resulting in $\eta$(298.15 K) = $8.9\cdot10^{-4}$ Pa·s and $\eta$(310.15 K) = $6.9\cdot10^{-4}$ Pa·s,

$$\eta_{water}(T) = A \cdot 10^{B/(T-C)}. \tag{7}$$

Using these viscosity values in **Equation 6** we derived that D(310.15 K) equals $5.51\cdot10^{-6}$ cm²/s and $5.49\cdot10^{-6}$ cm²/s for Oregon green and Atto 655-maleimide, respectively.

By fitting the obtained green and red auto-correlation curves to a diffusion model (**Equation 1**), the $\tau_D$s of Oregon green and of Atto 655-maleimide were found to be 16.4 ± 1.3 µs and = 24 ± 3.1 µs, respectively (mean and standard deviation values are derived from two independent measurement repeats). Unlike for $\tau_D$, the fitted values for the structural parameter, $s$, where highly variable and fixing it hardly affected the overall goodness of fit or $\tau_D$. Therefore, we instead derived $s$ from the auto-correlation curves of meGFP and TDmKate2, measured in cells expressing a TDmKate2-Don1-meGFP fusion protein (in which the Don1 serves as an inert linker to avoid FRET, **Maeder et al., 2007**). These measurements and fittings resulted in $s = 9.97 ± 0.06$ and $s = 9.98 ± 0.08$ for the green and red channels, respectively. Fixing these $s$ values (rounded to $s = 10$ for both channels) in the fitting of the Oregon green and of Atto 655-maleimide auto-correlation curves ended up with $\tau_D = 16.38 ± 1.23$ µs and $\tau_D = 23.34 ± 2.24$ µs, respectively. Based on these $\tau_D$ and $s$ values and **Equations 4**, **5** and **8**, we derived that $V_G = 0.38$ fl and $V_R = 0.67$ fl.

$$\omega_0 = \sqrt{\tau_D \cdot 4D}. \tag{8}$$

To quantify the effect of different maturation rates of TDmKate2- and meGFP we performed FCS measurements in cells expressing a TDmKate2-Don1-meGFP fusion protein (**Maeder et al., 2007**). In these measurements, the average ratio between the number of meGFP and TDmKate2 particles in the confocal volume ($N_{mKate2}^{Don1}/N_{meGFP}^{Don1}$), as derived from the green and red auto-correlation curves in several cells and in two sessions, was found to be 0.94 ± 0.21. However, since the green and red detection volumes are not the same, the correction ratio between the concentrations of TDmKate2 and meGFP is:

$$\varphi = \frac{N_{mKate2}^{Don1}}{N_{meGFP}^{Don1}} \cdot \frac{V_G}{V_R} = 0.94 \cdot \frac{0.38}{0.67} = 0.53. \tag{9}$$

This bias from the expected 1:1 ratio reflects, plausibly, the previously reported slow maturation rate of mKate2 (**Maeder et al., 2007**). Therefore, to compensate for the expected fraction of immature, dark, mKate2-labeled proteins in the confocal volume we need to divide $N_R^R$ and $N_{RG}^{RG}$ by this factor. Accordingly, **Equation 3** becomes:

$$Ka = \frac{N_{RG}^{RG}/(V_{RG} \cdot \varphi)}{\left\{ N_R^R/(V_R \cdot \varphi) - N_{RG}^{RG}/(V_{RG} \cdot \varphi) \right\} \left\{ N_G^G/V_G - N_{RG}^{RG}/(V_{RG} \cdot \varphi) \right\}}. \tag{10}$$

To derive $V_{RG}$ we performed FCCS measurements of $N_G$, $N_R$, and $N_{RG}$ in cells expressing the TDmKate2-Don1-meGFP fusion protein and solved for $V_{RG}^{Green}$ and $V_{RG}^{Red}$:

$$V_{RG}^{Green} = \frac{N_{RG} \cdot V_G/\varphi}{N_G^G} = 0.33 \text{ fl},$$

$$V_{RG}^{Red} = \frac{N_{RG} \cdot V_R/\varphi}{N_R^R/\varphi} = \frac{N_{RG} \cdot V_R}{N_R^R} = 0.355 \text{ fl}.$$

Averaging these values, we conclude that $V_{RG} = 0.34$ fl.

FCCS data of problematic measurements, due to cell movements and focus drifts, were automatically identified and excluded based on the following criteria: (i) if in one or two of the channels the obtained counts-per-molecule is ≤215 Hz, to ensure sufficient signal from each particle for detecting its thermal fluctuations in and out the confocal volume. (ii) If the offset in one or two of the channels is ≥0.0025 or ≤ −0.002, (iii) if $(C_G − C_D)/(C_R − C_D)$ ≥20, thereby ensuring (according to control measurements) that there is no significant bleedthrough from the green to the red channels, (iv) if the obtained $Ka$ is negative, as such a result indicates that the derived total protein concentrations are smaller than the derived protein-complex concentrations due to a too noisy cross-correlation curve.

## Statistical analysis

For each valid FCCS measurement (see above), an association score was derived from the cross-correlation curve by $score = \left( \tilde{x}_{amp} - \tilde{x}_{off} \right)/MAD(amp)$, where $MAD(amp)$ denotes the median of the absolute deviation from the median of the amplitude and $\tilde{x}_{amp}$ and $\tilde{x}_{off}$ denote the median of the amplitude and offset, respectively. The amplitude and offset of each curve were derived as described in the FCS and FCCS data analysis section. Quantification of the statistical confidence that a protein pair is physically associated was performed using Fisher's exact test comparing the distribution of the association scores of that protein pair with the distribution of the association scores of the negative control measurements. The null hypothesis, $H_0$, and the alternative, $H_1$, for this test were $H_0 : \tilde{\mu}_{pair} = \tilde{\mu}_{neg}$ and $H_1 : \tilde{\mu}_{pair} > \tilde{\mu}_{neg}$, where $\tilde{\mu}_{pair}$ and $\tilde{\mu}_{neg}$ denote the medians of the association scores of the protein pair and the negative control measurements, respectively. Accordingly, a lower obtained p-value indicates a higher confidence that the given protein pair is physically associated (*Supplementary file 1*, columns C–F). Fisher's exact test was chosen since it is suitable for small sample sizes and does not require any assumptions about the underlying family of compared distributions.

Quantification of the statistical significance of the overall difference in association scores near-versus-far from focal adhesions for all measured protein pairs together was obtained by testing them cell-wise coupled between the two locations using the nonparametric one-sample Wilcoxon signed rank test, because of non-normality of the data (*Supplementary file 1*, row 5, columns H–J). Here, the null hypothesis $H_0 : \tilde{\mu}_{S_1-S_2} = 0$ was tested against the alternatives $H_1 : \tilde{\mu}_{S_1-S_2} \neq 0$, $H_1 : \tilde{\mu}_{S_1-S_2} > 0$, and $H_1 : \tilde{\mu}_{S_1-S_2} < 0$, where $S_1$ and $S_2$ denote the vectors of the association scores near and far from focal adhesions, respectively. Accordingly, $\tilde{\mu}_{S_1-S_2}$ is the median of the differences of the association scores near-versus-far from focal adhesions over all measurements. The three alternative hypotheses were analyzed in order to cover the possibilities of any change, increase or decrease in the association score. Using the one-sample Wilcoxon signed rank test in the same manner, the statistical significance of the difference in association scores near-versus-far from focal adhesions for each individual protein pair was calculated (*Supplementary file 1*, rows 6–96, columns H–J).

As a complementary evaluation, we also quantified the statistical significance of the differences in association scores between the two conditions in an uncoupled manner (i.e., batching together measurements from different cells) for the protein pairs together and for the protein pairs individually. In parallel with the coupled analysis described above, this was done using the two-samples Wilcoxon signed rank test for the following null hypothesis $H_0 : \tilde{\mu}_1 - \tilde{\mu}_2 = 0$ against all possible

alternatives $H_1 : \tilde{\mu}_1 - \tilde{\mu}_2 \neq 0$, $H_1 : \tilde{\mu}_1 - \tilde{\mu}_2 > 0$, and $H_1 : \tilde{\mu}_1 - \tilde{\mu}_2 < 0$, where $\tilde{\mu}_1$ and $\tilde{\mu}_2$ denote the median of the association scores near and far from focal adhesions, respectively (*Supplementary file 1*, columns K–M). All statistical tests were performed using R (version 3.0.1, The R Foundation for Statistical Computing, Vienna, Austria; http://www.r-project.org).

## Plasmids

We constructed plasmids encoding adhesion site proteins tagged with either monomeric enhanced GFP (meGFP), a tandem of monomeric Kate2 (TDmKate2), mCitrine or mCherry (primers are described in *Supplementary file 2*):

### Backbone plasmids for meGFP-fusion

Backbone plasmids for meGFP-fusion were constructed on the basis of eGFP-N1 and eGFP-C1 (Clontech Laboratories Inc., Mountain View, CA) by introducing the A206K mutation by mutagenesis PCR using primers eGFP-meGFP-FP and eGFP-meGFP-RP.

### Backbone plasmids for TDmKate2-fusion

Backbone plasmids for TDmKate2-fusion were constructed based on mKate2-N1 and mKate2-C1 (the latter one was constructed by amplifying mKate2 from mKate2-N1 mKate2 FP1 and mKate2 RP2, and inserting it into peGFP via AgeI and XhoI sites). For the C1 backbone a PCR was conducted on mKate2-C1 using primers mKate2 C1 FP1 and mKate2 C1 RP1. The product was cut with AgeI and BspEI and inserted into mKate2-C1 backbone that was cut with BspEI. For the N1 backbone a PCR was conducted on mKate2-N1 using primers mKate2 N1 FP1 and mKate2 N1 RP1. The product was cut with AgeI and BspEI and inserted into mKate2-N1 backbone that was cut with AgeI. For the positive control TDmKate2-Don1-meGFP, the inert yeast protein Don1 was amplified from yeast genomic DNA with primers Don1-FP and Don1-RP and cut with BglII/EcoRI. SalI/BamHI restriction sites were introduced in meGFP using meGFP-N1 as template and meGFP-FP and meGFP-RP as primers. Both fragments were ligated into TDmKate2-C1.

### Backbone plasmids mCitrine-N1 and mCitrine-C1

Backbone plasmids mCitrine-N1 and mCitrine-C1 were kind gifts from Joel Swanson.

### Backbone plasmids mCherry-C1 and mCherry-N1

Backbone plasmids mCherry-C1 and mCherry-N1 were obtained from Clontech.

### α-Actinin constructs

pEGFP-N1-α-Actinin was obtained from Adgene. It was subcloned into meGFP-N1 and TDmKate2-N1 using the restriction sites EcoRI and XhoI.

### α-Parvin constructs

α-Parvin was amplified with primers Parvin C1 FP1 and Parvin C1 RP1 using HeLa cDNA as template. The product was cut with EcoRI and XhoI and inserted into meGFP-C1, mCitrine-C1, and TDmKate2-C1 backbones.

### CAS constructs

p130CAS was obtained from Openbiosystems (p130CAS-pOTB7). The linker was introduced by PCR using primers CAS C1 FP1 and CAS C1 RP1. The product was cut by XhoI/EcoRI and inserted into meGFP-C1, mCitrine-C1, and TDmKate2-C1 plasmids.

### CSK constructs

CSK was amplified with primers Csk N1 FP1 and Csk N1 RP1 using HeLa cDNA as PCR template. The product was cut with SalI and BamHI and inserted into meGFP-N1, mCitrine-C1, and TDmKate2-N1 backbones.

### FAK constructs

FAK was amplified using HeLa cDNA as PCR template. Restriction sites BglII and SalI were introduced using primers FAK C1 FP1 and FAK C1 RP1. The PCR product was cut and inserted into meGPF-C1, mCitrine-C1, and TDmKate2-C1 backbones.

### ILK constructs

ILK was amplified with primers ILK C1 FP1 and ILK C1 RP1 using HeLa cDNA as template. The product was cut with XhoI and EcoRI and inserted into meGFP-C1 and TDmKate2-C1 backbones.

### Paxillin constructs

mCherry-α-Paxillin-C1 was a kind gift of Irina Kaverina (*Efimov et al., 2008*). Paxillin was cut with BglII and EcoRI and subcloned into meGFP-C1, mCitrine, mCherry, and TDmKate2-C1.

### PINCH constructs

PINCH1 was obtained from Openbiosystems in pDNR-LIB (Catalog number MHS1011-62408). Linker and restrictions sites were introduced by PCR using primers PINCH-FP2 and PINCH-RP2, product was cut by BglII/SalI and inserted into meGFP-C1 and TDmKate2-C1 plasmids.

### Talin constructs

Talin was amplified by a 2-step RT-PCR using U373 mRNA as template and the primer Talin C1 RP2. After a PCR with primer Talin C1 FP2 and Talin C1 RP2 the product was cut in two parts using the internal restriction site FseI. The flanking restriction sites were EcoRI/FseI and FseI/SalI, respectively. Both parts were inserted in two steps into mCitrine-C1 that was modified with an additional FseI restriction site by mutagenesis PCR with primers mCitrine-FseI FP and mCitrine-FseI-RP. Talin was subcloned into meGFP-C1 and 2mKate2-C1 using EcoRI and SalI as restriction enzymes.

### Tensin constructs

eGFP-Tensin was a kind gift of David L Brautigan (*Hall et al., 2009*). Tensin was amplified using PCR with primers Tensin-FP and Tensin-RP. PCR product was cut by SalI/KpnI and ligated into meGFP-C1, mCherry, and TDmKate2-C1.

### VASP constructs

VASP was amplified with primers VASP N1 FP1 and VASP N1 RP1 using HeLa cDNA as template. The product was cut with EcoRI and BamHI and inserted into meGFP-N1, mCitrine-C1, and TDmKate2-N1 backbones.

### Vinculin constructs

mKO-Vinculin-C1 (kind gift from Miguel Vicente-Manzanares [*Choi et al., 2008*]) was used as PCR template for Vinculin. The restriction sites BspEI and SalI were introduced using primers Vinculin FP1 and Vinculin RP1. The PCR product was cut and inserted into meGFP-C1, mCherry, and TDmKate2-C1 backbones.

### Zyxin constructs

Cerulian-Zyxin-C1 was a kind gift of Irina Kaverina (*Efimov et al., 2008*). Zyxin was cut with EcoRI and BamHI and subcloned into meGFP-C1 and TDmKate2-C1.

### SH2 constructs

mCherry-dSH2-C1, based on YFP-dSH2 (*Kirchner et al., 2003*) (kind gift from Benjamin Geiger), was sub-cloned into meGFP-C1 with the restriction enzymes NheI and XhoI and adjusted to the right reading frame by mutagenesis PCR, using primers meGFP-dSH2 FP1 and RP1.

## Immunolabeling

For immunolabeling cells were washed with 37°C warm PBS, then incubated with freshly prepared 3% Paraformaldehyde solution for 10 min and subsequently permeabilized for 5 min by incubation with 0.2% Triton X-100 in PBS. Antibodies were diluted in PBS containing 0.2% Triton X-100. The fixed and permeabilized cells were incubated for 1 hr with phospho-specific primary antibody: rabbit IgG anti-FAK pY407 (Invitrogen catalog number 44-650G), rabbit IgG anti paxillin pY118 (Invitrogen catalog number 44-722G), or rabbit IgG anti p130CAS pY165 (Cell Signaling, catalog number 4015). Cells were then washed 3 times by incubation with PBS for 10 min each, subsequently incubated for 1 hr with the secondary antibody, goat anti-rabbit IgG conjugated with Alexa 405 (Invitrogen, catalog number A-31556), and washed with PBS before imaging.

## Acknowledgements

We thank Philippe Bastiaens and Benjamin Geiger for inspiring discussions, Lisaweta Roßmannek for help with plasmids construction, Martin Bierbaum for Matlab scripts, and Hernán Grecco and Astrid Krämer for helpful comments.

# Additional information

### Funding

| Funder | Grant reference number | Author |
| --- | --- | --- |
| The Ministry of Innovation, Science and Research of North Rhine-Westphalia | | Jan-Erik Hoffmann |
| Bundesministerium für Bildung und Forschung | | Jan-Erik Hoffmann |
| The International Leibniz Graduate School: Systems Biology Lab on a Chip | | Jan-Erik Hoffmann |
| The German Academic Exchange Service (DAAD) | | Yessica Fermin |
| Bundesministerium für Bildung und Forschung | 0315507 | Ruth LO Stricker, Katja Ickstadt, Eli Zamir |

The funders had no role in study design, data collection and interpretation, or the decision to submit the work for publication.

### Author contributions

J-EH, Conception and design, Acquisition of data, Analysis and interpretation of data, Designed and cloned the plasmids, Drafting or revising the article; YF, KI, Designed the statistical examination approach, Analysis and interpretation of data; RLOS, Designed the plasmid library cloning strategy, Designed and cloned the plasmids; EZ, Conception and design, Designed the statistical examination approach, Analysis and interpretation of data, Drafting or revising the article

# Additional files

### Supplementary files

• Supplementary file 1. **Statistical tests**. Columns A-B, the 91 protein pairs (arbitrarily ordered) and their index number. Columns C-F, number of measurements and the p-values derived from Fisher's Exact Tests comparing the distributions of association scores of the protein pairs (measured in REF52 or NIH3T3 cells) with the distributions of association scores of the negative control measurements (neg; meGFP and mKate2 in REF52). Columns G-M, number of measurements and the p-values derived from Wilcoxon Signed Rank Tests for the significance of the difference between the association scores near and far from focal adhesions for the various protein pairs. These differences were derived either by subtracting per-cell the association score measured near a focal adhesion (N) from that measured far from focal adhesions (F) (coupled; One-Sample Wilcoxon Signed Rank Tests) or by subtracting the median of the association scores measured near and far from focal adhesions over the same population of cells (uncoupled; Two-Sample Wilcoxon Signed Rank Tests). See 'Materials and methods' for further details.

• Supplementary file 2. **List of primers**. All primers are shown in a 5' to 3' orientation.

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
