## [Decision Letter]

Thank you for sending your work entitled “Symmetric and asymmetric exchange of multi-protein building blocks between focal adhesions and cytosol” for consideration at *eLife*. Your article has been favorably evaluated by a Senior editor and 2 reviewers, one of whom is a member of our Board of Reviewing Editors.

The Reviewing editor has assembled the following comments to help you prepare a revised submission.

While both reviewers found merit in the study, there were some concerns expressed. The first reviewer felt that the expression of the interacting proteins may be in excess and this would result in a skewed kinetic landscape. The second reviewer was very enthusiastic about the approach represented in the first three figures but objected to the use of drugs against motility in such high doses that the kinetics of disassembly would be non-physiological. It was suggested that the motility work be left out of the manuscript for this reason. The first reviewer concurs with this opinion.

We suggest that you revise the manuscripts to incorporate a response to these suggestions, and send it back for re-review.

*Reviewer*
*1:*

This work investigates the components of the adhesome in 91 pairwise combinations using FCCS, FLIM and FRAP, finding 15 that were significant in several cell lines. By measuring the association dynamics of these pairs with the focal adhesion, they can tease out specific “building blocks” of favored protein combinations and determine that their association and dissociation kinetics are symmetric – they go out of the complex with the same associations they go in with. The main conclusion is that the adhesome is assembled from multi protein building blocks and this facilitates the assembly process.

A lot of speculation then ensues about why this mode of assembly is good, for example rapid and robust responses. It's not clear why this would make such a big difference since the assembly of individual proteins into the complex would also be expected to be fast, and although this is a complex structure, it's not clear that complexity has to come from a subassembly route. Nevertheless, one wonders whether much has been gained conceptually by this work.

Furthermore, it is concerning that the components are expressed pairwise, probably in excess. In this case, if there is a high concentration of these components compared to endogenous levels, associations may be driven artificially by this overexpression, and form building blocks that would never have existed in vivo. This is particularly true if the adhesome, along with its endogenous assembly has saturated the components in the structural assembly. This is a significant flaw in the approach that I feel compromises the interpretation.

*Reviewer*
*2:*

This manuscript describes results of careful and elegant biophysical experiments designed to test the hypothesis that focal adhesions form and disassemble from pre-formed, specific macromolecular assemblies. Through thoughtfully designed experiments and brilliant analysis, the results reveal that specific preassembled building blocks incorporate into and disassemble from FAs symmetrically (i.e., without change in composition). Further, the results of their pairwise biophysical analysis reveal both known and new mutually exclusive assemblies, and suggests that specific preformed assemblies are recruited to each ultrastructural layer of the adhesion. In my 20 years of reviewing papers I have never had a paper where I wrote “beautiful” in the margin and “brilliant” after the concluding sentence of each paragraph of the results section so many times. This paper will provide a major, important advance in how we think about how focal adhesions are built.

In spite of my extreme enthusiasm for the first three figures, I do have a significant beef with interpretations of the experiments involving inhibition of contractility. These cells were hit with a hammer: 100uM of Y-drug for 30-60 minutes. This is problematic for two reasons. First, inhibition of contractility like this is not representative of the physiological mechanism of adhesion disassembly. Adhesions likely disassemble due to actions of calpain on FA proteins. Second, with this high concentration of drug, which is generally used an order of magnitude more dilute, for such a long treatment time, adhesions would be disassembled within 10 minutes. The cytosolic pool likely represents a new steady-state, not the state of recent post-disassembly, as conjectured by the authors. In addition, the length of treatment time also likely induces more plieotropic effects such as complete disassembly of stress fibers and changes in post-translational modification of all kinds proteins that are regulated by contractility-dependent mechano-signalling. I think to perform this experiment properly would require doing analysis at different time points after the addition of different concentrations of drug, finding the interesting candidates, and comparing this to what happens to these candidates in normal adhesions as they are disassembling by endogenously regulated pathways. In this Reviewer's opinion, that is beyond the scope of the current study, and may not even represent a physiologically relevant pathway. I feel that the myosin inhibition experiments could be removed and the figures re-tooled and the rest of the findings of the paper are such an important advance that the manuscript would be acceptable without expansion of the hard-to-interpret myosin inhibition study.

---

## [Author Response]

We have found the comments very helpful and revised the manuscript accordingly. The changes include: (1) Removing the data (former Figure 4) and text related to the actomyosin inhibition experiments, (2) Consequently, adding a paragraph discussing material exchange by disassembling focal adhesions, (3) Modifying the model (former Figure 5, currently Figure 4), (4) Adding a Figure supplement for Figure 1 and text addressing the issue of the ectopic expression levels, (5) Expanding the discussion in the text about the implications of the concepts and findings presented in this work, and (6) Adding the protein names in Figure 2 and other minor technical modifications.

Reviewer 1:

*This work investigates the components of the adhesome in 91 pairwise combinations using FCCS, FLIM and FRAP, finding 15 that were significant in several cell lines. By measuring the association dynamics of these pairs with the focal adhesion, they can tease out specific “building blocks” of favored protein combinations and determine that their association and dissociation kinetics are symmetric – they go out of the complex with the same associations they go in with. The main conclusion is that the adhesome is assembled from multi protein building blocks and this facilitates the assembly process*.

*A lot of speculation then ensues about why this mode of assembly is good, for example rapid and robust responses. It's not clear why this would make such a big difference since the assembly of individual proteins into the complex would also be expected to be fast, and although this is a complex structure, it's not clear that complexity has to come from a subassembly route. Nevertheless, one wonders whether much has been gained conceptually by this work*.

Since adhesion sites are self-assembled and heterogeneous structures, it is essential to reveal and consider the nature of their building blocks in order to understand how these sites are formed and regulated. So far this aspect was rather neglected and the cytosolic pool was tacitly conceived as a collection of individual proteins. Conceptually, this work implies that adhesion sites assembly is a process involving in multi-protein complexes as the elementary building blocks. Multi-protein building blocks reduce the number of steps that are needed for self-assembly of adhesion sites and therefore can make this process less prone to errors, thereby also faster. In comparison to a pool of single-protein building blocks, a pool of combinatorial diversified multi-protein building blocks also implies different regulatory principles for the self-assembly of adhesion sites. Therefore, the findings and conceptual novelty that we present in this work are important and relevant for broad aspects of adhesion sites formation and regulation. We learned from the comment that this aspect should be clarified in the text and accordingly we addressed it in a new paragraph added at the end of the Discussion section in the revised manuscript.

*Furthermore, it is concerning that the components are expressed pairwise, probably in excess. In this case, if there is a high concentration of these components compared to endogenous levels, associations may be driven artificially by this overexpression, and form building blocks that would never have existed in vivo. This is particularly true if the adhesome, along with its endogenous assembly has saturated the components in the structural assembly. This is a significant flaw in the approach that I feel compromises the interpretation*.

This is indeed an important point that requires a careful consideration. Here below we address the two aspects of (i) the actual levels of the ectopic expressions and (ii) their potency in forming new complexes in the cytosol context.

(i) The concentrations of the ectopically expressed proteins in the FCS/FCCS measurements are low: FCS/FCCS is a sensitive technique that measures concentrations and associations of labeled proteins at a nanomolar - micromolar concentration scale. Accordingly, in our measurements the concentrations of the ectopically expressed proteins in the cells (as we can quantify from the FCCS data) were almost all (98%) below 500 nM and the majority of them (84%) were also below 200 nM. Such a range of concentrations is comparable to the range of typical concentrations of endogenous proteins (10-1000 nM; [24], Cell 141:1262-1262 e1261). Therefore, in this work the ectopically expressed proteins are not in excess in comparison to the typical (though wide) range of endogenous protein concentrations.

(As a side comment: To the best of our knowledge and search, reported information about the absolute endogenous concentrations of each of the measured proteins in fibroblasts is very limited. Due to the sparse information, variability between methods, cell types and cell-volume estimations (Milo, 2013, Bioessays 35:1050-1055) we find it to be ambiguous to infer the endogenous concentrations of the 13 analyzed proteins in REF52 and NIH3T3 cells by integrating across current published information.)

(ii) The effect of ectopically expressed proteins on their complex concentration is buffered by competition with multiple other different endogenous proteins: Ectopic expression of two proteins (A and B) would lead to a formation of an [AB] complex that was practically absent before in the cytosol if it is in levels that can compete successfully with endogenous proteins that normally prevent the formation of [AB] (by competing with A on B and with B on A). Since each integrin adhesome protein typically has multiple different interactions (Zaidel-Bar, 2007, Nature Cell Biology 9:858-867), many of them mutually exclusive, the total concentration of all endogenous proteins competing on a given A-B interaction can be approximated as the average of typical endogenous concentrations in cells multiplied by the number of the different competing proteins. This total concentration is therefore considerably larger than the concentrations of the ectopically expressed proteins, as we showed that these ectopic expressions are within the typical range of single endogenous proteins. Obviously, not all copies of the competing endogenous proteins will be available for sequestering the ectopically expressed A and B proteins, as they will be also occupied by other interactions. Still, a given increase in the total concentrations of A and B will be distributed over the multiple different competing endogenous proteins. This considerably reduces the sensitivity of the [AB] complex concentration to a given elevation in total A and B concentrations.

The combination of this buffering effect and the low ectopic expression levels minimizes the alteration of the levels of the measured complexes in comparison to normal in-vivo conditions. Therefore, we expect that the complexes we detected exist in the cytosol of cells regardless of the ectopic expression.

In order to convey these points, we have added Figure 1—figure supplement 1,, showing the distribution of concentrations of the ectopically expressed proteins in the measurements used for Figure 1 and the range of typical concentrations of endogenous proteins. We summarized the abovementioned explanation in the legend of the figure supplement for Figure 1.

Reviewer 2:

*This manuscript describes results of careful and elegant biophysical experiments designed to test the hypothesis that focal adhesions form and disassemble from pre-formed, specific macromolecular assemblies. Through thoughtfully designed experiments and brilliant analysis, the results reveal that specific preassembled building blocks incorporate into and disassemble from FAs symmetrically (i.e., without change in composition). Further, the results of their pairwise biophysical analysis reveal both known and new mutually exclusive assemblies, and suggests that specific preformed assemblies are recruited to each ultrastructural layer of the adhesion. In my 20 years of reviewing papers I have never had a paper where I wrote “beautiful” in the margin and “brilliant” after the concluding sentence of each paragraph of the results section so many times. This paper will provide a major, important advance in how we think about how focal adhesions are built*.

*In spite of my extreme enthusiasm for the first three figures, I do have a significant beef with interpretations of the experiments involving inhibition of contractility. These cells were hit with a hammer: 100uM of Y-drug for 30-60 minutes. This is problematic for two reasons. First, inhibition of contractility like this is not representative of the physiological mechanism of adhesion disassembly. Adhesions likely disassemble due to actions of calpain on FA proteins. Second, with this high concentration of drug, which is generally used an order of magnitude more dilute, for such a long treatment time, adhesions would be disassembled within 10 minutes. The cytosolic pool likely represents a new steady-state, not the state of recent post-disassembly, as conjectured by the authors. In addition, the length of treatment time also likely induces more plieotropic effects such as complete disassembly of stress fibers and changes in post-translational modification of all kinds proteins that are regulated by contractility-dependent mechano-signalling. I think to perform this experiment properly would require doing analysis at different time points after the addition of different concentrations of drug, finding the interesting candidates, and comparing this to what happens to these candidates in normal adhesions as they are disassembling by endogenously regulated pathways. In this Reviewer's opinion, that is beyond the scope of the current study, and may not even represent a physiologically relevant pathway. I feel that the myosin inhibition experiments could be removed and the figures re-tooled and the rest of the findings of the paper are such an important advance that the manuscript would be acceptable without expansion of the hard-to-interpret myosin inhibition study*.

We agree with the reviewer that our finding that inhibition of actomyosin contractility increases physical associations between integrin adhesome components in the cytosol can be accounted for by different scenarios, as explained in the comment and indeed also in the manuscript. We initially included this part as it provides helpful information that can guide further exploration of the symmetry of material exchange by disassembling focal adhesions. However, following this comment, we agree with the view of the reviewer that the focus and strength of the manuscript will be benefitted from omitting the actomyosin inhibition part. Therefore, we revised the manuscript accordingly: (1) We removed the figure and text which are related to the Y-27632 experiments. (2) Since we still would like to mention and discuss the issue of material exchange by disassembling focal-adhesions, we have added a paragraph to the Discussion section.